# Towards Understanding the Interconnection between Celestial Pole Motion and Earth’s Magnetic Field Using Space Geodetic Techniques

**DOI:** 10.3390/s21227555

**Published:** 2021-11-13

**Authors:** Sadegh Modiri, Robert Heinkelmann, Santiago Belda, Zinovy Malkin, Mostafa Hoseini, Monika Korte, José M. Ferrándiz, Harald Schuh

**Affiliations:** 1Department Geodesy, Federal Agency for Cartography and Geodesy (BKG), 60322 Frankfurt am Main, Germany; 2GFZ German Research Centre for Geosciences, 14473 Potsdam, Germany; robert.heinkelmann@gfz-potsdam.de (R.H.); monika.korte@gfz-potsdam.de (M.K.); schuh@gfz-potsdam.de (H.S.); 3UAVAC, University of Alicante, 03080 Alicante, Spain; santiago.belda@uv.es (S.B.); jm.ferrandiz@ua.es (J.M.F.); 4Image Processing Laboratory (IPL)-Laboratory of Earth Observation (LEO), University of Valencia, 46980 Valencia, Spain; 5Central Astronomical Observatory at Pulkovo, 196140 St. Petersburg, Russia; malkin@gaoran.ru; 6Department of Civil and Environmental Engineering, Norwegian University of Science and Technology, 7491 Trondheim, Norway; mostafa.hoseini@ntnu.no; 7Institute for Geodesy and Geoinformation Science, Technische Universität Berlin, 10623 Berlin, Germany

**Keywords:** celestial pole offset, geomagnetic field, VLBI

## Abstract

The understanding of forced temporal variations in celestial pole motion (CPM) could bring us significantly closer to meeting the accuracy goals pursued by the Global Geodetic Observing System (GGOS) of the International Association of Geodesy (IAG), i.e., 1 mm accuracy and 0.1 mm/year stability on global scales in terms of the Earth orientation parameters. Besides astronomical forcing, CPM excitation depends on the processes in the fluid core and the core–mantle boundary. The same processes are responsible for the variations in the geomagnetic field (GMF). Several investigations were conducted during the last decade to find a possible interconnection of GMF changes with the length of day (LOD) variations. However, less attention was paid to the interdependence of the GMF changes and the CPM variations. This study uses the celestial pole offsets (CPO) time series obtained from very long baseline interferometry (VLBI) observations and data such as spherical harmonic coefficients, geomagnetic jerk, and magnetic field dipole moment from a state-of-the-art geomagnetic field model to explore the correlation between them. In this study, we use wavelet coherence analysis to compute the correspondence between the two non-stationary time series in the time–frequency domain. Our preliminary results reveal interesting common features in the CPM and GMF variations, which show the potential to improve the understanding of the GMF’s contribution to the Earth’s rotation. Special attention is given to the corresponding signal between FCN and GMF and potential time lags between geomagnetic jerks and rotational variations.

## 1. Introduction

The Earth’s rotation can provide essential information regarding the Earth system as several processes contribute to its excitation from the inner part of the Earth to the outer layers. Therefore, the Earth’s rotation time series have generated great interest in different fields in geoscience and astronomy [1,2,3]. Earth orientation parameters (EOP) are the five angles that show the Earth’s surface orientation in space, and they are used to relate points in the terrestrial and celestial reference system. The celestial intermediate pole (CIP) variation in the terrestrial frame is described by polar motion. UT1-UTC is a linear function of the Earth’s rotation angle, which gives the distance between terrestrial and celestial origins on the common intermediate equator. Finally, the CIP variation in the celestial frame is called celestial pole motion (CPM). The IAU2006 precession [4,5] and IAU2000A nutation models [6] were adopted to provide accurate approximations and predictions of the CIP. However, they are not complete and not fully accurate, and very long baseline interferometry (VLBI) observations show that the CIP deviates from the position resulting from the application of the IAU2006/2000A model (see, e.g., [7]). These deviations or offsets of the CIP are known as celestial pole offsets (CPO) and are denoted as (dX, dY). Currently, accurate observations of CPO can only be obtained by the VLBI technique. The observed CPO can quantify the deficiencies of the IAU2006/2000A precession–nutation model, including the astronomically forced nutations and a component of nutation that is considered unpredictable. The latter is mainly constituted by the free core nutation (FCN), which is excited by angular momentum exchanges between the Earth’s mantle and its fluid layers [8,9]. It has a retrograde period of approximately 430 days (with average amplitudes of around 100 µas) relative to the inertial frame [10], or a period slightly shorter than 1 day in the retrograde diurnal band, relative to the rotating terrestrial frame. Although the FCN is a geodynamic effect, according to the current definition, it is expressed with regard to the celestial system. Finally, it should be noted that the current theories and models cannot predict the actual Earth’s rotation with the accuracy corresponding to the current observations and prospective needs. There are several reasons for this, such as imperfection of physical models, inconsistency between terrestrial and celestial reference systems and frames, systematic errors, and unmodeled geophysical signals [11,12,13,14,15,16].

Belda et al. [17] determined a new set of empirical corrections to the precession offsets and rates and re-assessed the amplitudes of a broad set of terms included in the IAU 2006/2000A precession–nutation theory. Applying these corrections to the CPO, some signals, e.g., at 1024 days, were found in the remaining residuals. The signals could be caused by different geophysical phenomena, such as strong El $\tilde{N}$ino southern oscillation (ENSO), free inner core nutation [6,18,19,20], or geomagnetic field (GMF) variation [21,22]. During the last decade, several investigations have been performed to discuss a possible interconnection of GMF changes with the Earth’s rotation parameters, such as polar motion and length of day (LOD) [23,24,25,26,27,28,29]. However, less attention has been paid to the celestial part, including the impact of the GMF changes, such as the geomagnetic jerks (GMJ), which are rapid changes in GMF secular variations, and other GMF variations on the CPM fluctuations, such as the free mode of the Earth’s rotation caused by the Earth’s core and mantle’s different material characteristics, i.e., FCN [8,9,30]. Shirai et al. [21] studied the association between the GMJ and FCN, which revealed a close coincidence of two FCN phase jumps with two GMJs that occurred in 1992 and 1999, which was confirmed by other studies [22,31]. Malkin [20,22] showed that the extreme variations in the amplitude and phase of FCN coincide with GMJ epochs. This means that the FCN can be excited by the same processes that cause the GMJ, which could be close to reality, as the flows in the core mainly generate the GMJ, and the same flows lead to variations in the core moments of inertia and thus can cause the FCN variations [32].

Moreover, some studies have been conducted to determine nutations considering the magnetic field’s influence mathematically [33,34,35]. The amplitudes of nutation are calculated in a displacement field method, incorporating a prescribed magnetic field inside the Earth’s core. A magnetic field’s shearing is caused by relative motion between the liquid core and outer solid parts. After this, an incremental magnetic field is generated, which returns and perturbs the nutations themselves, as is addressed in a nutation model estimated from an angular momentum budget method [36,37,38].

In addition, Huang et al. [38] proposed a new strategy to estimate nutations, considering the magnetic field’s influence directly in the motion equation and in the boundary condition. Their results indicate that the FCN period decreases by 0.38 days, and the out-of-phase (in-phase) amplitudes of the -18.6-year and the -1-year nutations increase (decrease) by 20 and 39 µas, respectively.

Although there is some indication for this correlation, some doubts about the connection structure between the FCN and the Earth’s surface geophysical fluids still need to be explored (see [31,39,40]). In addition, the distribution of GMJ events should be considered since they are registered by geomagnetic observatories (as well as satellite observations), and they can be observed worldwide (global jerks) or within specific geographic regions (regional jerks).

In this study, the relationships between FCN and GMJ, magnetic dipole moment (DM), and GMF models are investigated to determine the dependency structure between the CPM and GMF. Our results confirm the assumptions of previous studies [13]. The paper is structured as follows. The data set and theory about the FCN and GMF are described in Section 2. Section 3 outlines the applied methods. A discussion on the workflow’s strengths and weaknesses and future research lines is presented in Section 4, followed by the conclusions, presented in Section 5.

## 2. Data Set and Time Series Analysis

A brief description of the data sets used in this study is given as follows.

### 2.1. GMF Data

Here, the geomagnetic field model, GMJ, and magnetic dipole moment time series, which are used in this study, will be expressed.

#### 2.1.1. Geomagnetic Field Model

The GMF data used in this study were obtained from the CHAOS model series [41,42,43,44]. The model aims to represent the internal GMF at the Earth’s surface with high resolution in space and time. The CHAOS core field provides information on the temporal variations in the core of the main part of the Earth’s magnetic field [45]. The CHAOS model is derived primarily from magnetic satellite data, as well as monthly observatory mean data. The CHAOS model series consider vector measurements at mid- to low-latitudes and scalar data. The CHAOS’s validity is restricted to post-1999, when the Orset satellite was launched.

In this study, the CHAOS-6 core field model is used to investigate the dependency structure between the GMF data and the FCN time series. The CHAOS-6 model parametrization follows that of CHAOS-5 and CHAOS-4. See Olsen et al. [44] for a more detailed account of the CHAOS field modeling scheme, including the external model. The time-dependent internal field $B_{int}\left(t\right)=-\nabla V_{int}$ is designed as the gradient of the scalar potential:(1)$$V_{int}\left(r,\theta ,\phi \right)=a\sum_{n=1}^{10}\sum_{m=0}^{n}\left[g_{n}^{m}\left(t\right)\cos m\phi +h_{n}^{m}\left(t\right)\sin m\phi \right]{\left(\frac{a}{r}\right)}^{n+1}P_{n}^{m}\left(\cos\theta \right)$$
where a=6371.2 km is a reference radius, (r,θ,ϕ) are geographical coordinates, and $P_{n}^{m}\left(\cos\theta \right)$ are the Schmidt semi-normalized associated Legendre functions of degree *n* and order m. $g_{n}^{m}\left(t\right)$ and $h_{n}^{m}\left(t\right)$ are time-dependent Gauss coefficients.

The description in terms of spherical harmonic coefficients (SHC) with constant Gauss coefficients is applicable for stationary snapshot fields. However, since the magnetic field changes in space and time, all field models use time-dependent Gauss coefficients. The most commonly adopted approach is to model the temporal evolution with spline functions [46]. The SHC are widely used to describe the magnetic field as a derivative of potential in a spherical coordinate system. This represents the magnetic field potential as a series of multipoles, where n=1 represents the dipole contribution, and n=2 represents the quadrupole contribution. In this study, the SHC up to degree and order ten are taken from the CHAOS-6 model as these coefficients are more related to the core activity [47]. As the GMJ events are detectable in the secular variation (SV) of the geomagnetic field, we investigate the rate of change of SV, estimated by the second derivative of GMF. The SHC can be grouped into three main categories: zonal (e.g., $C_{i0}$, i=1,…,10), tesseral (e.g., $C_{ij}$, i,j≠0, and i≠j), and sectoral ( $C_{ii}$ or $S_{ii}$, i=1,…,10). Figure 1 shows the normalized power spectra of the second derivative of zonal, tesseral, and sectoral SHC up to degree 2. The spectra were calculated by applying a multivariate Least-Squares Harmonic Estimation (LSHE) analysis [48] to multiple coefficient time series of the same type to retrieve common mode signals between several time series at once. The figure reveals common periods among different types of harmonics, e.g., the period of around 30 months, which is common among all. Hence, spectral analysis was performed to explore the dominant recurring terms.

#### 2.1.2. GMJ

GMJs occurred mainly in 1999, 2003/2004, 2007/2008, 2011, and 2014 within our study period [49,50,51,52]. Besides these confirmed jerks, Sabaka et al. [53] derived a comprehensive model (CM4) of the geomagnetic field based on hourly data of magnetic observatories and satellites and found another jerk of questionable global extent in 1997. Malkin [20] also considered a possible jerk in 1994 based on some observations at geomagnetic stations. Because satellite data have made it possible to calculate the global secular acceleration (SA) of the geomagnetic field, some large SA pulses at the core surface were found in 2006, 2009, and 2012.5, which may be related to GMJs observed at the Earth’s surface [54].

#### 2.1.3. Magnetic Dipole Moment

The Earth’s main dipolar magnetic (DM) field results from the convective movement of the electrically conducting fluid iron–nickel mix that forms the liquid outer core at depths between roughly 3480 and 5150 km. This field is tilted by approximately $10.5^{\circ }$ from the Earth’s rotational axis and varies smoothly in space and time due to changing current interactions in the core. The DM is calculated from SHC models using the first three coefficients as follows [55]:(2)$$\mathrm{DM}=\frac{4\pi }{\mu_{0}}R^{3}\sqrt{{\left(g_{1}^{0}\right)}^{2}+{\left(g_{1}^{1}\right)}^{2}+{\left(h_{1}^{1}\right)}^{2}}$$
where R=6371.2 km, and $\mu_{0}=4\pi \times 10^{-7}$ Vs/(Am) the permeability of free space. Figure 2 shows the DM rate and the second derivative of DM obtained from the CHAOS6 model. The DM rate increased from mid-1999 to 2003. The d^2^DM/dt^2^ changed between 2002 and 2005 very sharply. As demonstrated in Figure 2, the minima and maxima of the DM rate time series occurred coincidentally at GMJ events. Another notable feature is that the d^2^DM/dt^2^ shows significant steady changes during the GMJ events.

Table 1 presents the results of the LSHE analysis of the zonal, sectoral, and tesseral parts of GMF’s second derivative up to degree 2 and DM (rate and second derivative). The dominant signals in the GMF’s SHC parameters are relative with a period of 7, 4, and 3 years. Moreover, some signals with a period of 20–33 months are found in GMF’s SHC and DM rate and its second derivative, which are in common with FCN signals.

### 2.2. FCN Data

Belda et al. [56] developed a new empirical FCN model (named B16) with a high temporal resolution by fitting the amplitude parameters directly to the celestial pole offsets (CPO) solution calculated by VLBI.

FCN models can be characterized by a weighted least-squares fit of these equations [57]:(3)$$FCN_{X}=A_{C}cos\left(\sigma_{FCN}t\right)-A_{S}sin\left(\sigma_{FCN}t\right)+X_{0}$$
$$FCN_{Y}=A_{S}cos\left(\sigma_{FCN}t\right)-A_{C}sin\left(\sigma_{FCN}t\right)+Y_{0}$$
where $\sigma_{FCN}=\frac{2\pi }{P}$ is the frequency of FCN in the CRF, $A_{c}$ and $A_{S}$ correspond to sine and cosine amplitudes, t is the time relative to J2000.0, *P* is the period, and $X_{0}$ and $Y_{0}$ are constant offsets. These offsets incorporate the low-frequency part of the signal. Therefore, the contribution of the FCN to the CPO can be computed by using Equation (Equation 3) without taking into account the constant offsets: $X_{0}$ and $Y_{0}$.

Figure 3 illustrates CPO observations obtained from VLBI using the Potsdam Open-Source Radio Interferometry Tool (PORT) [58] and the emprical B16 FCN model estimated between 1990 and 2017. The B16 model is fitted to VLBI data using a sliding window with a length of 400 days, a step size of one day, and a fixed period of −431.18 sidereal days.

Since this study relies upon the dynamic behavior of the signal, observed and modeled FCN data are transformed from the data space into the normalized space between 0 and 1. Figure 4 shows the normalized FCN amplitude, offset, and phase variations and their first derivatives with respect to time, estimated by using the B16 FCN model and Equations (3) and (4) reported in [59]. Note that the FCN amplitudes show a general long-time decrease before 1999. They subsequently grow until 2011 and then seem to decrease again. On the other hand, similar FCN phase behavior could also be observed, i.e., the long-time FCN phase drift changed suddenly in 1998–1999. According to different studies [40,60,61,62,63], this could be interpreted as a FCN frequency change because the FCN phase and period variations cannot be distinguished from the mathematical processing of the FCN time series. However, including other observation data, such as absolute gravimetry, that show that the FCN period is quite stable, and taking into account the theoretical considerations, allows us to clarify this point and conclude that the observed FCN variations are mainly explained by the phase change over time.

In Figure 4, the red dashed boxes represent the GMJs. The green boxes indicate a significant magnetic secular acceleration pulse (SA) at the core mantle boundary. The yellow boxes show uncertain GMJ events [20]. After down-sampling of the FCN parameters from daily to monthly resolution, the spectral analyses of the FCN parameters and their rate were performed using the fast Fourier transform (FFT) (Table 2 and Table 3). The results show several remarkable features. Signals with periods of 23–33 months could be detected in most FCN components from 1990 to 2017. The $FCN_{X}$, $FCN_{Y}$, $FCN_{Phase}$, and $FCN_{offset}$ rate contain signals with a period of 33.2 months.

We applied a change detection method [64] based on Singular Spectrum Analysis (SSA) [65,66] for the simultaneous detection of amplitude and phase-induced changes. Originally, the SSA-based method was introduced to reveal abrupt hardware- or software-related changes in the atmospheric data products of Global Navigation Satellite Systems (GNSS). Here, the method was used to examine significant changes within the CPO and LOD time series for possible timely interconnection to the geomagnetic events, including GMJs and SA pulses. The output of the SSA-based method is an empirical index called the Change Magnitude Estimator (CME), which is a quantifier for the change events and their relative magnitudes. To estimate the position and magnitude of a change point, the method first creates a multi-dimensional vector space. The orthogonal basis function of this linear space is derived from the elements of the time series. Then, the derived basis function is used to extract a representative trend of the time series, which can be considered a smoothed version of the time series. Finally, the distribution of the residuals, i.e., the time series after subtracting the trend, is used to calculate the CME index. More details of this change detection method can be found in [64].

The time index and magnitude of the CME values (Figure 5) indicate the estimates of the epoch and significance of the change points. In Figure 5, the CME graphs are overlaid on the reported GMJ events and SA pulses for visual comparison. The time series of the CME index for LOD (Figure 5—right panel) shows overall very good agreement with the SA pulses (blue-shaded highlights), except for the one in 2009, which exhibits a relatively insignificant peak. Similar agreement between the CPO changes and the SA pulses can be seen in the left panel of the figure, except for the SA event in 2006, which could not show similar significance to the other SA pulses.

The GMJ in 1999 was concurrent with a sharp change indication in the CME time series of CPO and with an apparent delay in the LOD time series. In 2003/2004, the CME index had relatively high values in the CPO time series. This event was noticeably detected in the LOD time series as well. In both time series, the peaks coinciding with the reported GMJ in 2007/2008 and 2011 were small. The GMJ event in 2014 could be related to two peaks in the CPO time series and, with a delay of around one year, to a prominent peak in the LOD time series, considering that magnetic activity was not the only possible effect.

## 3. Method

In this study, we employed the wavelet coherence analysis (WCA) method to analyze the time series’s coherence as a function in both domains, time and frequency. The WCA is based on the continuous wavelet transform (CWT), which is defined as:(4)$$W^{X}\left(n,s\right)=\sqrt{\frac{\Delta t}{s}}\sum_{\overset{‵}{n}=1}^{N}x\left(n\right)\psi_{0}^{⋆}\left[\overset{‵}{n}-n\right]\left(\frac{\Delta t}{t}\right)]$$
where *W* denotes the CWT of a time series x(n), *n* the time index, *s* the wavelet scale, *N* the length of the time series, Δt the time step, $\psi_{0}$ the mother wavelet function, and ^⋆^ indicates the complex conjugate.

The cross-wavelet power spectrum between x(n) and y(n) is defined as in [67]:(5)$$W^{XY}\left(n,s\right)=W^{X}\left(n,s\right)W^{Y*}\left(n,s\right)$$
where the $W^{XY}\left(n,s\right)$ denotes the joint power between x(n) and y(n) and * indicates the complex conjugate again.

Furthermore, the squared cross-wavelet coherence function $R^{2}$ is used to describe how coherent the cross-wavelet transform is in the time–frequency domain as follows [68,69]:(6)$$R^{2}\left(n,s\right)=\frac{|S\left(s^{-1}W^{XY}\left(n,s\right)\right){|}^{2}}{S(s^{-1}|W^{X}{\left(n,s\right)|}^{2})\times S(s^{-1}|W^{Y}\left(n,s\right){|}^{2})}$$
where *S* is a smoothing operator. Equation Equation 6 resembles the traditional correlation coefficient, and it is helpful to think of the wavelet coherence as a localized correlation coefficient in the time–frequency domain. The smooth operator *S* is written as [69]:(7)$$S\left(W\right)=S_{scale}\left(S_{time}\left(W_{n}\left(s\right)\right)\right),$$
where $S_{scale}$ denotes smoothing along the wavelet scale axis and $S_{time}$ smoothing in time. The statistical significance level of the wavelet coherence is computed by using Monte Carlo methods.

## 4. Discussion of Results

The association between FCN and GMJ, magnetic dipole moment, and GMF models was investigated to explore the dependency structure between the CPM and GMF time series using wavelet coherence analysis.

### 4.1. Geomagnetic Spherical Harmonic Coefficients and FCN

The coherence analysis between the FCN (amplitude, phase, offset, and their rate) and the spherical harmonic coefficients (SHC) up to degree and order ten was examined. First, the WCA between all individual SHC and FCN parameters was computed. Then, the principal component analysis (PCA) was performed on the WCA results to extract the main coherence components between FCN and GMF’s SHC. For this purpose, we embedded each WCA result as one column of a matrix. The PCA was then applied to the constructed matrix. The statistically significant long-term coherence was clearly distinguished in most cases, more concretely at time scales of 24–36 months. The upper panel of Figure 6 shows the wavelet coherence analysis between the second derivative of each coefficient of GMF’s spherical harmonic and $FCN_{Amplitude}$ rate. The wavelet coherence values close to high correlation are displayed in red, whereas the blue color shows low or no correlation between parameters. The upper panel shows the percentage variability explained by each principal component. In the upper right panel, the coherence analysis reconstructed by all principal components (PC) is displayed. Several locally significant coherences varying from 4 to 12 months, as well as a long-term correlation with a period of approximately 16 and 32 months, could be identified. In the middle panel, the coherence analysis shows the features of the first PC (PC = 1), which demonstrates weak long-term coherence in approximately the 32-month band, which is almost confirmed by more than 65% of the studied coefficients. However, the lower panel shows statistically significant long-term coherence at approximately 24–36 months. The higher correlation with lower-degree and -order coefficients shows that they have greater sensitivity to the Earth’s core activities. In addition, the results of WCA between GMF’s SHC and $FCN_{Phase}$ rate are shown in Figure 7. The upper panel shows that several local coherences coincide approximately at the GMJ’s epochs, e.g., 1999, 2004, 2008, and 2014 in the 12-month band, which is the same as the assumed duration of a GMJ event. Moreover, statistically significant long-term coherence can be seen in the middle panel, confirming that more than 65% of the coefficients had sensitivity to GMJ events. The lower panel demonstrates the very significant correlation between lower SHC and $FCN_{Phase}$ at the period between 24 and 36 months during the whole time period.

As an example, the result of WCA of $S_{22}$ and $FCN_{Phase}$ is shown in Figure 8. The results demonstrate statistically significant long-term coherence at periods between 24 and 36 months over the whole interval of time, and several local coherences can also be identified in the 10-to-16-month band.

### 4.2. FCN and Magnetic Dipole Moment

The association between FCN parameters and DM variation was investigated in daily temporal resolution. The WCA indicated a statistically significant long-term coherence extending from 1997 to 2018. In particular, the approximate period of 1024 days (32 months) could be identified in most cases between FCN components and DM (first and second derivatives). As shown in Figure 9, some statistically significant local coherence was also found when GMJ events were reported. The upper plot shows the coherence between the DM rate and $FCN_{Amplitude}$ rate. There was significant long-term coherence in the 1024-day band. Although the correlation was weaker before 2003, it showed a very strong correlation in later years. The lower plot shows episodic terms, occurring during almost one year at epochs that coincided with GMJ events.

## 5. Conclusions

The primary purpose of this paper was a further investigation of the connection between the CPM and GMF to extend our understanding of the Earth’s rotation and improve the Earth’s rotation theory. Furthermore, the primary practical goal of this study was the improvement of the CPM prediction accuracy. Thus, a better understanding of CPM excitation could bring us significantly closer to meeting the accuracy goals pursued by the GGOS of the IAG. Although both FCN and GMF depend on the processes in the Earth’s core and at the core–mantle boundary effects, no physical mechanism has been identified yet to explain how GMF can influence FCN. However, many previous studies discussed above showed clear manifestations of the GMF variations, particularly GMJs, in the EOP variations, such as LOD and FCN. In this study, we performed a more detailed investigation of the temporal coherence between GMF on the CPO. The CPO time series were obtained from VLBI observations and the latest GMF data to explore the association between CPO and the GMF using the wavelet coherence analysis technique. Our results confirm previous studies’ results by indicating that substantial FCN amplitude and phase disturbances occurred at the epochs close to the revealed GMJ events [20,22,40]. Our results also revealed some common features in the FCN and GMF variation, which show the potential to improve the knowledge regarding the GMF’s contribution to the Earth’s rotation. The shown results are consistent with the conclusion of Gibert et al. [70], who found that the rapid changes in the Chandler Wobble follow the GMJ with a delay of 1 to 3 years. Moreover, the WCA of the GMF’s elements and FCN identify the coherence of around 32 months (approximately 1024 days), which confirms the findings of Belda et al. [13]. They suggested that the periodic signal near 1024 days in FCN offsets could be caused by the GMF variation and its sudden changes. The lower degree of SHC describes the core activity, and our investigation could be related to the mentioned fact. Therefore, the lower SHCs were investigated because of their direct relationship with the Earth’s core activity. The coherence analysis between the FCN (offsets, amplitude, and phase) and SHC rate indicated a statistically significant long-term coherence with periods of 3–5, 18–24, and 36 months over the whole interval of time. The WCA of DM and FCN indicates a significant association between the DM and FCN, which could work as a potential external parameter to improve the FCN prediction by considering DM information. WCA detects the coherence between the offset of FCN rate and DM rate at 32 months (approximately 1024 days), as well as statistically significant local coherence around 1999, 2003, and 2007 within a band of four months when the GMJ happened. Although our results cannot contribute to establishing the physical relationship between the FCN and GMF, they still demonstrate the mathematically significant coherence between both phenomena. As not all GMJs occur along with FCN variations and vice versa, our next step will be to investigate and identify the properties that establish this coherence. A starting point could be the spatial extension of the GMJs. Furthermore, we will consider extending the study period towards including more recent events and explore ways to work with increased temporal resolution in order to identify a potential time delay between GMF and FCN variations. Therefore, GMJ, DM, and GMF model study can potentially improve our understanding of FCN excitation mechanisms. 

## Figures and Tables

**Figure 1 sensors-21-07555-f001:**
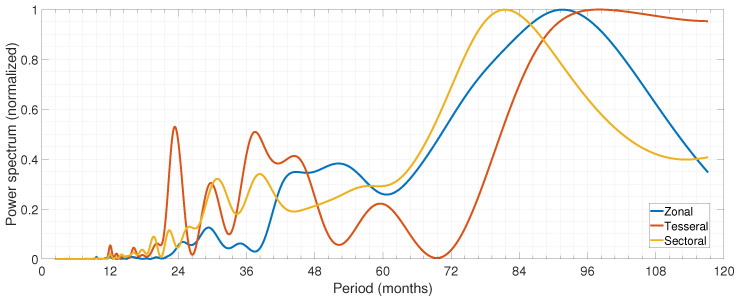
Multivariate spectral analysis of the second derivative of zonal, tesseral, and sectoral spherical harmonics coefficients up to degree 2.

**Figure 2 sensors-21-07555-f002:**
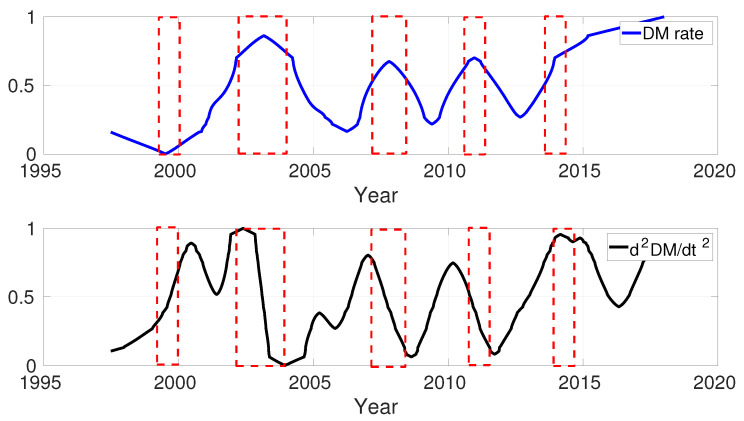
Time series of dDM/dt and d^2^DM/dt^2^ obtained from CHAOS6 model.

**Figure 3 sensors-21-07555-f003:**
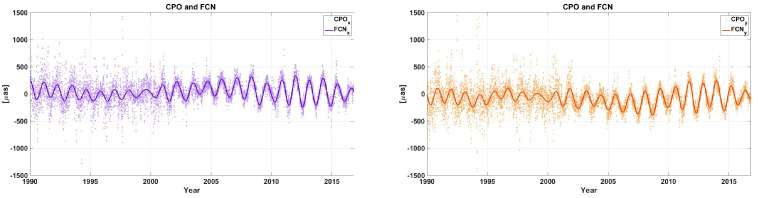
CPOs (dots) and FCN model B16 (line) in X (purple) and Y (orange) direction.

**Figure 4 sensors-21-07555-f004:**
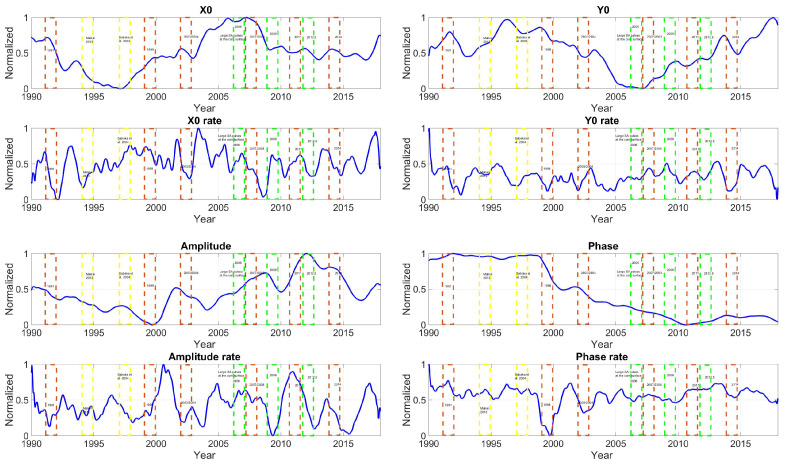
Time series of the normalized FCN offset, amplitude, and phase for the B16 model. The dashed box indicates GMJ and magnetic secular acceleration pulse (SA) at the core surface. The red color shows the confirmed GMJ. The yellow shows the questionable SA of the GMJ. The green indicates the significant global SA of the GMF.

**Figure 5 sensors-21-07555-f005:**
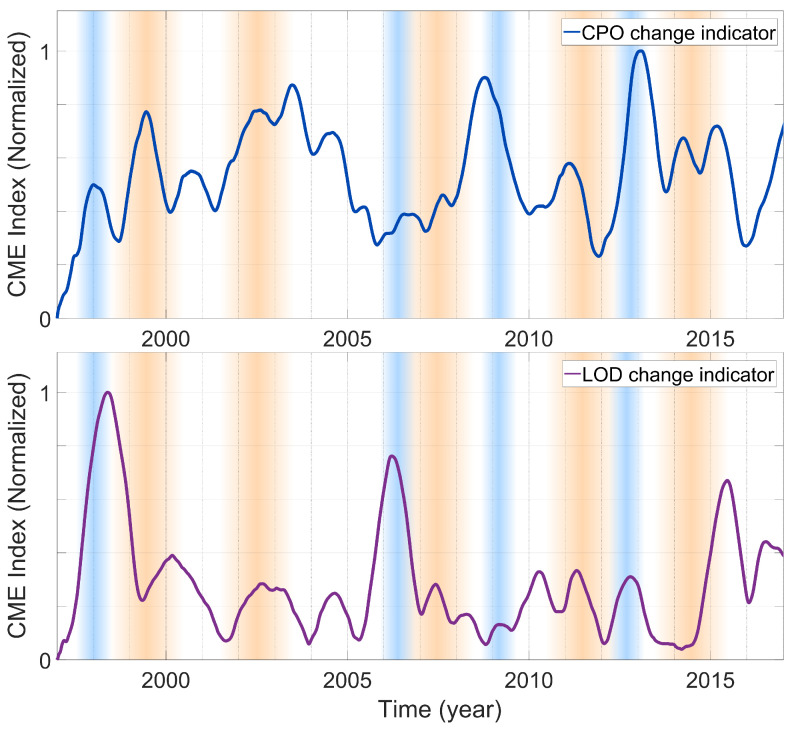
Detection of the time and significance of changes in the CPO (top) and LOD (bottom) time series based on Singular Spectrum Analysis (SSA). The blue-shaded areas indicate the epochs of SA pulses and the orange-shaded areas show reported geomagnetic jerks.

**Figure 6 sensors-21-07555-f006:**
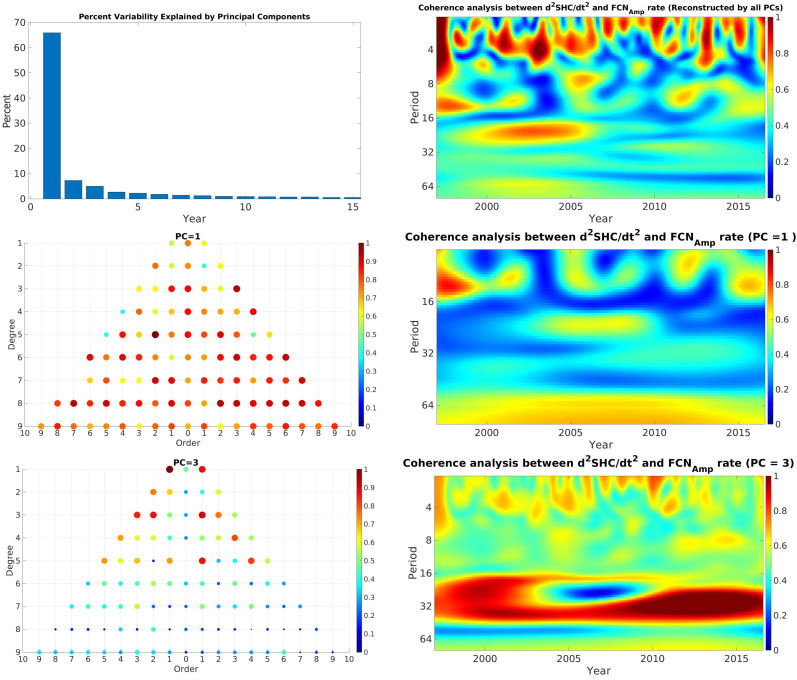
Wavelet coherence analysis between all SHC of GMF and $FCN_{Amp}$ rate (upper panel) and the percentage of each PC. Middle left panel: SHC correlation with the first PC. Middle right panel: the coherence between GMF and $FCN_{Amp}$ rate reconstructed by PC = 1. Lower left panel: SHC correlation with the PC = 3. Lower right panel: the coherence between GMF and $FCN_{Amp}$ rate reconstructed by PC = 3. Unit of periods is month.

**Figure 7 sensors-21-07555-f007:**
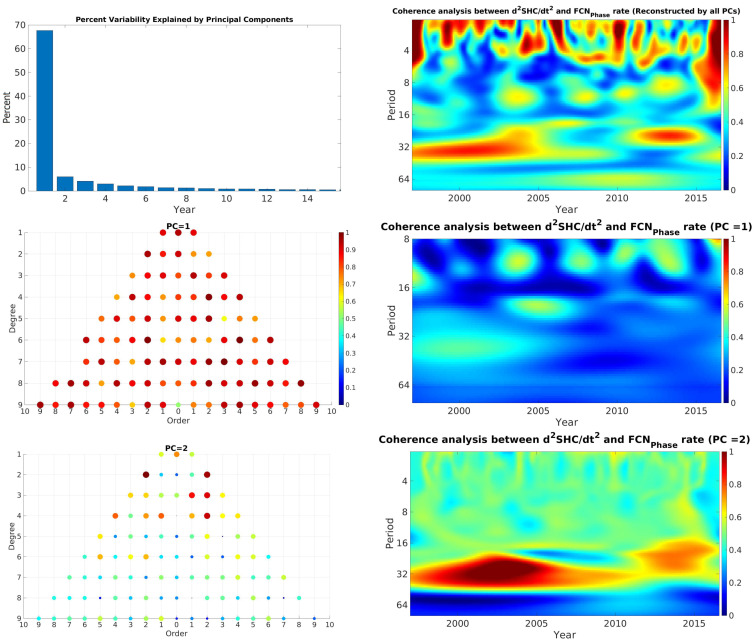
Wavelet coherence analysis between all SHC of GMF and $FCN_{Phase}$ rate (upper panel) and the percentage of each PC. Middle left panel: SHC correlation with the first PC. Middle right panel: the coherence between GMF and $FCN_{Phase}$ rate reconstructed by PC = 1. Lower left panel: SHC correlation with the PC = 2. Lower right panel: the coherence between GMF and $FCN_{Phase}$ rate reconstructed by PC = 2. Unit of periods is month.

**Figure 8 sensors-21-07555-f008:**
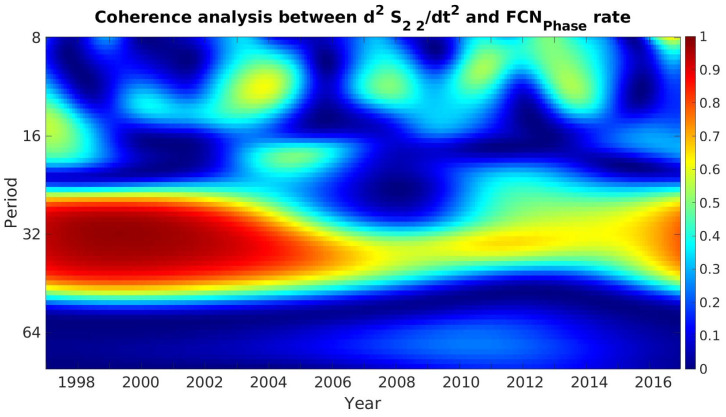
Wavelet coherence analysis between $\frac{d^{2}S_{2\prime \prime }}{dt^{2}}$ and $FCN_{Phase}$ rate.

**Figure 9 sensors-21-07555-f009:**
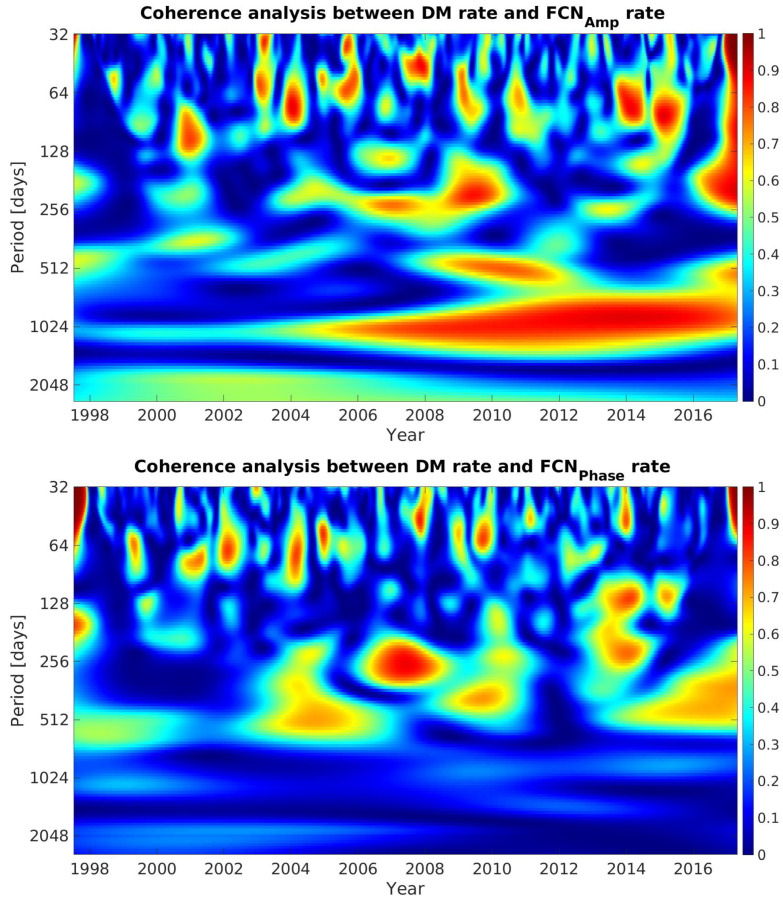
Wavelet coherence between FCN (amplitude and phase) rate and DM rate.

**Table 1 sensors-21-07555-t001:** Periods of the second derivative of GMF’s SHC up to degree 2 and DM (rate and second derivative) time series (months) obtained with spectral analysis.

	GMF_*Zonal*_	GMF_*Tesseral*_	GMF_*Sectoral*_	DM Rate	d^2^DM/dt^2^
1	91.7	97.8	81.6	81.3	83.1
2	52.2	23.4	38.2	40.5	49.9
3	44.3	37.5	30.9	24.3	31.1
4	29.4	44.5	25.6	17.3	24.9
5	24.8	29.7	22.4	1.9	15.5

**Table 2 sensors-21-07555-t002:** Spectral analysis of the FCN time series (after changing the resolution from daily to monthly): first five dominant frequencies (unit: months).

	*FCN_X_*	*FCN_Y_*	*FCN_X_0__*	*FCN_Y_0__*	*FCN_Amp_*	*FCN_Phase_*
1	15.1	15.1	27.7	1.66	33.2	55.3
2	33.2	55.3	18.4	23.7	23.7	33.2
3	20.8	33.2	12.8	16.6	15.1	18.4
4	8.7	11.1	10.8	13.8	11.1	11.9
5	7.5	23.7	8.7	11.9	9.2	15.1

**Table 3 sensors-21-07555-t003:** Spectral analysis of the first derivative of FCN time series (after changing the resolution from daily to monthly): first five dominant frequencies (unit: months).

	*FCN_X_*	*FCN_Y_*	*FCN_X_0__*	*FCN_Y_0__*	*FCN_Amp_*	*FCN_Phase_*
1	15.1	15.1	83	166.0	15.1	15.1
2	11.1	27.7	23.7	23.7	23.7	11.9
3	8.7	41.5	33.2	55.3	11.9	55.3
4	27.7	8.7	15.1	33.2	8.7	8.7
5	41.5	7.5	10.4	8.7	83.0	18.4

## Data Availability

Data is available upon request to correspondence author.

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
