# Peer review of "Towards Understanding the Interconnection between Celestial Pole Motion and Earth’s Magnetic Field Using Space Geodetic Techniques"

_sensors, 2021, doi:10.3390/s21227555_

Round 1

Reviewer 1 Report

The article by S. Modiri et al. is dedicated to FCN signal variations and possible interpretation by geomagnetic jerks. It is well written. But to my opinion, the presented results do not really show evidently the correspondence of FCN amplitude and phase to geomagnetic jerks.

 Another question – what is the input of the coauthors into this article? For example, the FCN model of  S. Belda is used, but model of Z. Malkin – not compared with it. J. Ferrandish is well-known specialist in nutations, Hamiltonian formalism, etc., but I do not see ‘his hand’ in the text. The fact, that coauthors wrote papers in this field and are cited, does not automatically clarify their contribution to this presented article.

I also have several notes, some of them are small, some are fundamental:

Line 21 orientation on space,   may be in space

Line 24 similarly - on the celestial frame

Line 29 – precession and nutation is only part of Earth rotation, defined as low-frequency (below 0.5 cpd) changes of the pole position in celestial frame. Thus it cannot fully explain Earth rotation in principle.

Line 69 what means amplitudes of the retrograde by 18.6 years  ?

Line 61 – you do not quite explain physics of FCN and CPO offsets. Surface geophysical fluids can only excite FCN under some hypothesis, see for example A. Brzezinsky et al.

Line 104 – it seems that in figure 1 you average very different spherical coefficients, zonal, sectorial, tesseral – is there any justification for their averaging? For example, zonal coefficients of different degrees are very different; do they represent any similar behavior to be averaged? I can see annual cycle in the curves, does not it complicate the understanding?  If you use unsmoothed curves, derivative and especially second derivative will be very sawtoothed, because numerical derivative calculation is an ill-posed procedure. It would be good to regularize this operation by smoothing, and/or mention which formula you use.  Roughness of the curves of amplitude and phase rate do not really allow to see much of correspondence between them and geomagnetic jerks.

Line 121 You do not present any equations how to calculate DM.

Line 144 similarly, how do you calculate CPO amplitude and phase? In 400-day sliding window?  Would be good to explain the procedure.

Figure  4 – it seems, that your curves can be smoothed in many different ways. The drift of phase after 2000 can be interpreted as FCN frequency change, see V.S. Gubanov papers (I recommend also to cite Younghon Zhou and Ben Chao’s papers on FCN, Dehant and Mathews book on Earth rotation, 2015).

Table 2,3 – The results of Fourier transform are usually shown in form of amplitude spectrum, but you give harmonics in the tables. How do you sort the presented frequencies – by their amplitude? In this case different periods appear in different rows. Won’t it be better to show plots, then we will see correspondence between FCN frequencies.

Equation 4 – is there multiplication in denominator? Last comma looks like conjugation?

 Figure 4 – what is shown with tiny letters on the plots? Could you mention in the caption, which is shown by red rectangles, which – by green.

Figure 5 – sorry, but I do not really see very good agreement with SA pulses. For example, year 2016 was very important for LOD, it started to decrease after it. Somewhere pulses are in advance, somewhere after. Can you explain, how you find correspondence? You use data up to 2021 here, while everywhere else – only up to ~2017. There were no GMJ after 2015?

Line 155 – I would recommend putting more references to SSA, such as M. Ghil, M. Allen et al.  Golyandina et al.  etc. May be more explanation on CME is needed.

Line 156 – was introduceD

In section 4 – do you present discussion on overall results or on wavelets only?

Line 186 – How do you apply PCA to WCA?

Line 215 FCNAmplitude – without space, cursive

Line 235 How “our investigation proves” exactly, that lower degree SHC describes the core activity?

Line 238 – what are these numbers 3o5…

Figure 6, 7 – where is % , top left y axis says period. Why you do not put shadow on the edge effects? It would be good to show grid lines for GMJ dates.

Figure 8 – is not it better to write years horizontally without inclination?

Figure 9 – if we look at the curves of DM rate Fig 2 top and amplitude and phase rate Fig 4 bottom do we really see any similarity? I agree, that wavelet coherence (see also A. Lyubushin) can reveal some frequencies, where there will be energy in both signals, but besides wavelet-coherence, there are also simple covariance function and Fourier cross-spectrum, why you do not study them? There should be some sense of soil underfoot interpretations of coherence. The final statement should not be “pulled by the ears”.

Literature is hard to access in non-alphabetic order.

My opinion is that the paper is not yet ready for publication in such a high-level journal as Sensors. It needs at least major revision. Though the style of the research and explanations is very good and the main author for sure has good methodological abilities, but the conclusions should be reinforced, or, at least, discussed persuasively.

Author Response

Response to Reviewer 1 Comments:

The article by S. Modiri et al. is dedicated to FCN signal variations and possible interpretation by geomagnetic jerks. It is well written. But to my opinion, the presented results do not really show evidently the correspondence of FCN amplitude and phase to geomagnetic jerks.

Response:

First of all, we greatly appreciate the time and effort you took to read our manuscript in detail that helped improve the quality of the paper. We did our best to meet all of the points and comments you raised in the revised version.

Please notice that an idea about the connection between the GMF variations, including the GMJs, and variations of the EOP has been considered by many authors for many years. The present paper hopefully contributes to these efforts considering in more detail the connection between GMF and CPM. However, we do agree with the reviewer that this topic still demands more research. Therefore, we have modified the title to clarify that this study is part of a more prominent framework. Consequently, we modified the article title to "Towards Understanding The Interconnection Between Celestial Pole Motion and Earth's Magnetic Field using Space Geodetic Techniques", as it would be our first approach to study GMF and FCN relationship.

Another question – what is the input of the coauthors into this article? For example, the FCN model of S. Belda is used, but model of Z. Malkin – not compared with it. J. Ferrandish is well-known specialist in nutations, Hamiltonian formalism, etc., but I do not see ‘his hand’ in the text. The fact, that coauthors wrote papers in this field and are cited, does not automatically clarify their contribution to this presented article.

Response:

Concerning your comment, we clarified the contribution to the article as follows:

Author Contributions

Conceptualization, Sadegh Modiri, Robert Heinkelmann, Santiago Belda, Zinovy Malkin, Mostafa Hoseini, Monika Korte, José M. Ferrándiz and Harald Schuh; Data curation, Sadegh Modiri, Santiago Belda and Mostafa Hoseini; Formal analysis, Sadegh Modiri, Santiago Belda and Mostafa Hoseini; Funding acquisition, Harald Schuh; Investigation, Sadegh Modiri, Santiago Belda and Mostafa Hoseini; Methodology, Sadegh Modiri, Santiago Belda and Mostafa Hoseini; Resources, Harald Schuh; Software, Sadegh Modiri and Mostafa Hoseini; Supervision, Robert Heinkelmann, José M. Ferrándiz and Harald Schuh; Validation, Sadegh Modiri, Robert Heinkelmann, Mostafa Hoseini, Monika Korte and José M. Ferrándiz; Visualization, Sadegh Modiri, Santiago Belda and Mostafa Hoseini; Writing – original draft, Sadegh Modiri, Robert Heinkelmann, Santiago Belda, Zinovy Malkin, Mostafa Hoseini, Monika Korte, José M. Ferrándiz and Harald Schuh; Writing – review & editing, Sadegh Modiri, Robert Heinkelmann, Santiago Belda, Zinovy Malkin, Mostafa Hoseini, Monika Korte, José M. Ferrándiz and Harald Schuh.

I also have several notes, some of them are small, some are fundamental.

Line 21 orientation on space,   may be in space

Response:

Done!

Line 24 similarly - on the celestial frame

Response:

Done!

Line 29 – precession and nutation is only part of Earth rotation, defined as low-frequency (below 0.5 cpd) changes of the pole position in celestial frame. Thus it cannot fully explain Earth rotation in principle.

Response

We specify the phrase according to your note.

“Finally, it should be noted that the current theories and models cannot predict the Earth's rotation with the accuracy corresponding to the current observations and prospective needs. There are several reasons for that such as imperfection of physical models, inconsistency between terrestrial and celestial reference systems and frames, systematic errors, and unmodeled geophysical signals”

Line 69 what means amplitudes of the retrograde by 18.6 years?

Response:

We replaced the word retrograde by using plus signs and minus signs and modified the text. Plus signs and minus signs denote prograde and retrograde, respectively. i.e.:  −18.6 year, −1 year, +18.6 year and +1 year

This sentence has now been amended for better clarity:

“The Free Core Nutation (FCN) period decreases by 0.38 day, and the out‐of‐phase (in‐phase) amplitudes of the −18.6 year and the −1 year nutations increase (decrease) by 20 and 39 μas, respectively.”

Line 61 – you do not quite explain physics of FCN and CPO offsets. Surface geophysical fluids can only excite FCN under some hypothesis, see for example A. Brzezinsky et al.

Response:

Following the reviewer’s suggestion, we have updated the article explaining the physics of FCN and CPO as follows:

“The IAU2000A nutation (Mathews et al. 2002) and IAU2006 precession models (Capitaine et al. 2003, 2005) were adopted to provide accurate approximations and predictions of the CIP. However, they are not fully accurate and VLBI (Very Long Baseline Interferometry) observations show that the CIP deviates from the position resulting from the application of the IAU2006/2000A model (see e.g., Petit & Luzum 2010). Those deviations or offsets of the CIP are known as Celestial Pole Offsets (CPOs) and are denoted as (dX, dY). Currently, accurate observations of CPO can only be obtained by the VLBI technique. The observed CPO can quantify the deficiencies of the IAU2006/2000A precession-nutation model, including the astronomically forced nutations and a component of nutation that is considered unpredictable. The latter is mainly constituted by the free core nutation (FCN), which is excited by angular momentum exchanges between the Earthʼs mantle and its fluid layers (Toomre 1974; Smith 1977). It has a retrograde period of about 430 days (with average amplitudes of about 100 μas) relative to the inertial frame (Krásná et al. 2013), or a period slightly shorter than 1 day in the retrograde-diurnal band, relative to the rotating terrestrial frame.”

Line 104 – it seems that in figure 1 you average very different spherical coefficients, zonal, sectorial, tesseral – is there any justification for their averaging? For example, zonal coefficients of different degrees are very different; do they represent any similar behavior to be averaged? I can see annual cycle in the curves, does not it complicate the understanding?  If you use unsmoothed curves, derivative and especially second derivative will be very sawtoothed, because numerical derivative calculation is an ill-posed procedure. It would be good to regularize this operation by smoothing, and/or mention which formula you use.  Roughness of the curves of amplitude and phase rate do not really allow to see much of correspondence between them and geomagnetic jerks.

Response:  

Regarding Figure 1, we investigate the potential of studying the relationship between the magnetic field in spherical harmonic models and the FCN. However, the variations of the GMF are slow, and the GMJ events occur within one to two years. Therefore, we have considered some assumptions here to investigate the possibility of matching GMJs with FCN. 

However, in Figure 1, we have interpreted the coefficients by dividing them into three main categories: zonal, tesseral, and sectoral. We have anticipated the most straightforward linear combination (mean) between each group. Next, the first derivative of each data set is taken. Then the data were smoothed using moving average windows. Next, the second derivative is calculated from the smoothed first derivative.

Finally, the data is transformed from data space to normalized space since we are interested in the dynamic behavior of time series.

In the end, we would like to confirm your concerns about the characteristics of each coefficient, and we have addressed this point in Figures 6 and 7. 

Line 121 You do not present any equations how to calculate DM.

Response: 

we added the DM equation as follow: 

\textcolor{red}{The DM is calculated from SHC models using the first three coefficients as follows \citep{merrill96}:

\begin{equation}

M=\frac{4 \pi}{\mu_0}R^{3}\sqrt{(g_{1}^{0})^2+(g_{1}^{1})^2+(h_{1}^{1})^2}  

\end{equation}

where, $R=6371.2$ km, and $\mu_0=4\pi.10^{-7}$ Vs/(Am) the permeability of free space.}

Line 144 similarly, how do you calculate CPO amplitude and phase? In 400-day sliding window?  Would be good to explain the procedure.

 Response:

The FCN amplitude and phase were estimated using a sliding window size of 400 days with a minimal displacement between the subsequent fit (one-day step). For this, we used our B16 FCN model B16 (Belda et al. 2016). The model is actually explained in the text.

To take into account the reviewer’s suggestion, we modified the text of the corresponding paragraph explaining the procedure as follows:

“Figure 4 shows the normalized FCN amplitude, offset and phase variations and their first derivatives with respect to time, estimated by using the B16 FCN model and the equations 3 and 4 reported in [Malkin 2007].”

Malkin, Z.M.  Empiric models of the Earth’s free core nutation.Solar System Research 2007,41, 492–497.390 doi: https://doi.org/10.1134/S0038094607060044 

Figure  4 – it seems, that your curves can be smoothed in many different ways. The drift of phase after 2000 can be interpreted as FCN frequency change, see V.S. Gubanov papers (I recommend also to cite Younghon Zhou and Ben Chao’s papers on FCN, Dehant and Mathews book on Earth rotation, 2015).

 Response:

 Thanks for the point. We have improved the text based on this comment and added the mentioned references.

Note that the FCN amplitudes showed a general long-time decrease before 1999. It subsequently grew until 2011 and then seemed to decrease again. On the other hand, a similar FCN phase behaviour can also be observed, i.e. the long-time FCN phase drift changed suddenly in 1998–1999. According to different studies [Dehant and Mathews 2015, Cui et al., 2020, Chao 2017, Gubanov 2010, Gubanov 2009], it could be interpreted as FCN frequency change because the FCN phase and periods variations cannot be distinguished from mathematical processing of the FCN time series. However, including other observation data such as absolute gravimetry that show that the FCN period is quite stable, and taking into account the theoretical considerations, allows us to clarify this point and conclude that observed FCN variations are mainly explained by the phase change over time.

New references inserted:

Dehant, V.; Mathews, P.M. Precession, Nutation and Wobble of the Earth; Cambridge University Press, 2015. doi:10.1017/CBO9781316136133

Cui, X.; Sun, H.; Xu, J.; Zhou, J.; Chen, X. Relationship between free core nutation and geomagnetic jerks. Journal of Geodesy 2020, 94.

Chao, B.F. On rotational normal modes of the Earth: Resonance, excitation, convolution, deconvolution and all that. Geodesy and Geodynamics 2017, 8, 371–376. Geodesy, Astronomy and Geophysics in Earth Rotation, doi:https://doi.org/10.1016/j.geog.2017.03.014.

Gubanov, V.S. New estimates of retrograde free core nutation parameters. Astronomy Letters 2010, 419 36, 444–451.

Gubanov, V.S. Dynamics of the Earth’s core from VLBI observations. Astronomy Letters 2009, 35, 270–277. 421.

Table 2,3 – The results of Fourier transform are usually shown in form of amplitude spectrum, but you give harmonics in the tables. How do you sort the presented frequencies – by their amplitude? In this case different periods appear in different rows. Won’t it be better to show plots, then we will see correspondence between FCN frequencies.

 Response:

Good point. Our approach: First the FFT is estimated, and then the dominating signals are sorted based on their magnitude. We think that Table 1 and Table 2 give the reader a good overview of the used dataset by knowing the detailed information.

Including the spectral analysis figures would mean that we would have to add 10 figures (5 figures for the FCN time series and 5 for the derivative of FCN time series) to the paper. We have used two tables to avoid including this high number of figures and at the same time, the table summarizes the most important information on the outcome of the FFT analysis.

Equation 4 – is there multiplication in denominator? Last comma looks like conjugation?

 Response:

It is added “(*) indicates the complex conjugate.”

Figure 4 – what is shown with tiny letters on the plots? Could you mention in the caption, which is shown by red rectangles, which – by green.

 Response:

We added the explanations into the caption.

Figure 5 – sorry, but I do not really see very good agreement with SA pulses. For example, year 2016 was very important for LOD, it started to decrease after it. Somewhere pulses are in advance, somewhere after. Can you explain, how you find correspondence? You use data up to 2021 here, while everywhere else – only up to ~2017. There were no GMJ after 2015?

 Response:

 The point raised by the reviewer is true. However, the occurrence of SA pulses and GMJs can reasonably precede or succeed other connected geophysical parameters such as LOD or CPO. Moreover, the accuracy of detection for the time indices of changes can be a few months (Hoseini et al. 2019). Therefore, slight shifts between the detected changes and the reported GMJs or SA pulses, which themselves have some uncertainties in terms of occurrence time, would be expected. Regarding the prominent change detected in the LOD time series, we have a preceding reported GMJ which might be one of the origins of  the detected LOD change Besides, the GMJs are normally confirmed in several years after the event. For instance, e.g.,  Kotze (2020), discusses the GMJs around 2015.5 and 2016.5, i.e. in the interval of your interest. 

Line 155 – I would recommend putting more references to SSA, such as M. Ghil, M. Allen et al.  Golyandina et al.  etc. May be more explanation on CME is needed.

 Response: 

Done!

Line 156 – was introduceD

 Response:

Done!

 In section 4 – do you present discussion on overall results or on wavelets only?

 Response:

The results are mainly based on wavelet analysis because it is one of the novel elements of our study w.r.t. Previous ones.. However, we consider other studies’ results for comparison. In the future we will consider new techniques, such as machine learning to find clear evidence that proves what we found by Wavelet analysis.

Line 186 – How do you apply PCA to WCA?

 Response:

We have added the following explanation in response to this comment:

For this purpose we have embedded each WCA result as one column of a matrix. The PCA is then applied to the constructed matrix. 

Line 215 FCNAmplitude – without space, cursive

 Response:

Done!

Line 235 How “our investigation proves” exactly, that lower degree SHC describes the core activity?

 Response:

We modified the text as follows:

 The lower degree of SHC describes the core activity, and our investigation \textcolor{red}{could be related to the} mentioned fact.

Line 238 – what are these numbers 3o5…

Response:
It was a mistake. Corrected: 3–5, 18–24, and 36 

Figure 6, 7 – where is % , top left y axis says period. Why you do not put shadow on the edge effects? It would be good to show grid lines for GMJ dates.

Response:
It is corrected as the y axis is percentage.

Figure 8 – is not it better to write years horizontally without inclination?

Response:

You are right. we modified the figure as you suggested.

Figure 9 – if we look at the curves of DM rate Fig 2 top and amplitude and phase rate Fig 4 bottom do we really see any similarity? I agree, that wavelet coherence (see also A. Lyubushin) can reveal some frequencies, where there will be energy in both signals, but besides wavelet-coherence, there are also simple covariance function and Fourier cross-spectrum, why you do not study them? There should be some sense of soil underfoot interpretations of coherence. 

Response:

The visual inspection would not clearly detect all the local or global signals in the time series. That’s why we have selected an advanced spectral analysis, wavelet coherence analysis, to reveal local or global correspondence of two different time series. The usage of techniques such as FFT means we miss some local patterns in expense of finding more accurate global frequencies.

 The final statement should not be “pulled by the ears”. Literature is hard to access in non-alphabetic order.

Response:
I agree with your point. I favor alphabetic order instead of sorting by numbers, but it was requested by Sensors journal, and we need to follow their rules.

My opinion is that the paper is not yet ready for publication in such a high-level journal as Sensors. It needs at least major revision. Though the style of the research and explanations is very good and the main author for sure has good methodological abilities, the conclusions should be reinforced, or, at least, discussed persuasively.

Response: 

We made our best to meet all of the points and comments that you raised in the revised version. We hope the revised version is now suitable for publication and look forward to hearing from you in due course.

Reviewer 2 Report

This study aims to study the connection between the CPM and GMF, which may help to improve the CPM prediction and thus to meet the accuracy pursued by the GGOS of the IAG. By comparing CPO and GMF data using the wavelet coherency analysis, the authors confirmed that the substantial FCN amplitude and phase disturbance occurred almost at the same time for GMJ events, though the physical mechanism is still unclear. In general, I think this study may be accepted for publication provided the following concerns are properly handled.

  1. There are so many abbreviations that the readers may not follow them easily. At least part of the abbreviations are not necessary.
  2. The GMJ events are too limited (only 5) to reach a sound conclusion (at least 1 or 2 out of 5 events do not show clear agreements between GMJs and CPOs according to Figs.1, 2, 4 and 5). It seems that the GMJs occurred every 3~4 years. Besides the mentioned 1994, 1997, 1999, 2003/2004, 2007/2008, 2011, and 2014, perhaps the authors can find more GMJs around 2017 and 2020from the new CHAOS-7.8 geomagnetic field model(spanning 1999-2021.4), and thus show more clues.
  3. The abstract needs a revision to stress the method(s) used and the conclusions obtained in this study. The sentences that are not quite relevant can be moved to the Introduction.
  4. There are quite a few typos in the Abstract and main texts, such as “to meeting (Lines 2 and others)”, “the geomagnetic field model, GMJ (Line 75)”, “a period of 3 ̆5, 18 ̆24, and 36 (Line 238)” and so on.

Author Response

Response to Reviewer 2 Comments:

This study aims to study the connection between the CPM and GMF, which may help to improve the CPM prediction and thus to meet the accuracy pursued by the GGOS of the IAG. By comparing CPO and GMF data using the wavelet coherency analysis, the authors confirmed that the substantial FCN amplitude and phase disturbance occurred almost at the same time for GMJ events, though the physical mechanism is still unclear. In general, I think this study may be accepted for publication provided the following concerns are properly handled.

Response:

We greatly appreciate the time and effort you took to read our manuscript in detail and for your very detailed and constructive comments that greatly helped improving the quality of the paper. We did our best to meet all the points and comments you raised in the revised version. Considering your general statement, we considered your concern about the physical mechanism, and then we modified our article’s title. 

We offer detailed responses to your comments as below:

Point 1: There are so many abbreviations that the readers may not follow them easily. At least part of the abbreviations are not necessary.

Response:

We removed some abbreviations in order to improve readability.

Point 2: The GMJ events are too limited (only 5) to reach a sound conclusion (at least 1 or 2 out of 5 events do not show clear agreements between GMJs and CPOs according to Figs.1, 2, 4 and 5). It seems that the GMJs occurred every 3~4 years. Besides the mentioned 1994, 1997, 1999, 2003/2004, 2007/2008, 2011, and 2014, perhaps the authors can find more GMJs around 2017 and 2020from the new CHAOS-7.8 geomagnetic field model(spanning 1999-2021.4), and thus show more clues.

 Response: 

Of course, one could include the more recent GMJ events into the analyses as well. Our results, however,  are based on the GMJ until 2014. As the time allowed for revision is very short (a few days only), we cannot extend the work beyond its current state. Please mind that the primary purpose of the paper is to verify a hypothesis mentioned in the manuscript already as a conclusion: "Although both FCN and GMF depend on the fluid core processes and the core-mantle boundary, no physical mechanism is identified to explain how GMF can influence FCN." This hypothesis does not necessarily require the involvement of the newest GMJ.

 Point 3: The abstract needs a revision to stress the method(s) used and the conclusions obtained in this study. The sentences that are not quite relevant can be moved to the Introduction.

 Response: 

 We revised the abstract by considering your suggestion.

Point 4: There are quite a few typos in the Abstract and main texts, such as “to meeting (Lines 2 and others)”, “the geomagnetic field model, GMJ (Line 75)”, “a period of 3 ̆5, 18 ̆24, and 36 (Line 238)” and so on.

 Response:

Throughout the text we fixed all the remaining typos.

Round 2

Reviewer 1 Report

it is ok now